# The Effect of Er:YAG Laser on a Shear Bond Strength Value of Orthodontic Brackets to Enamel—A Preliminary Study

**DOI:** 10.3390/ma14092093

**Published:** 2021-04-21

**Authors:** Jan Kiryk, Jacek Matys, Anna Nikodem, Karolina Burzyńska, Kinga Grzech-Leśniak, Marzena Dominiak, Maciej Dobrzyński

**Affiliations:** 1Oral Surgery Department, Wrocław Medical University, Krakowska st. 26, 50-425 Wrocław, Poland; jan.kiryk@umed.wroc.pl (J.K.); kgl@periocare.pl (K.G.-L.); marzena.dominiak@umed.wroc.pl (M.D.); 2Department of Mechanics, Materials and Biomedical Engineering, Wroclaw University of Science and Technology, Wybrzeże Wyspiańskiego st. 27, 50-370 Wrocław, Poland; anna.nikodem@pwr.edu.pl (A.N.); karolina.burzynska@pwr.edu.pl (K.B.); 3Department of Periodontics, School of Dentistry Virginia Commonwealth University, 12th St, Richmond, VA 23298, USA; 4Department of Pediatric Dentistry and Preclinical Dentistry, Wrocław Medical University, Krakowska st. 26, 50-425 Wrocław, Poland; maciej.dobrzynski@umed.wroc.pl

**Keywords:** microtomography, orthodontic brackets, roughness, shear strength, surface conditioning

## Abstract

We sought to evaluate the effects of Er:YAG laser (LightTouch, LightInstruments, Israel) conditioning on enamel roughness and shear bond strength of orthodontic brackets on enamel. Eighteen human molars (n = 9) and premolars (n = 9), were divided into 3 groups depending on the enamel conditioning method; Er:YAG laser (G1, n = 6), conventional etching with 37% orthophosphoric acid (G2, n = 6), Er:YAG laser combined with conventional etching (G3, n = 6). Er:YAG laser parameters were as follows: energy: 100 mJ, frequency: 10 Hz, exposure time: 10 s, applicator diameter: 600 μm, fluence: 35.37 J/cm^2^, distance: 1 mm away from a tooth, cooling: 80%. An MTS 858 MiniBionix^®^ machine was used to determine the shear bond strength (MTS System, Eden Prairie, MN, USA). The enamel structure was assessed using X-ray microtomography (SkyScan 1172, Bruker, Kontich, Belgium). The highest values of shear bond strength were obtained in the G3 group (9.23 ± 2.38 MPa) and the lowest values in the G2 group (6.44 ± 2.11 MPa) (*p* < 0.05). A significant change in the enamel surface was noted after applying laser, reaching up to 9% of enamel thickness, which was not observed in the etched samples. Moreover, the Er:YAG laser-irradiated enamel surface was characterized by the greatest roughness. The combined use of an Er:YAG laser with a conventional etching improves the adhesion of composite materials to the tooth.

## 1. Introduction

Minimally invasive dentistry (MID) aims to preserve as much tooth structure as possible to obtain the largest possible surface area for restorative material attachment without dental pulp damage. An important element of obtaining a high shear bond strength of dental restorative materials to enamel is an appropriate micromechanical and chemical preparation of the tooth surface. The traditional technique of composite material application involves preparing the enamel surface by etching it with 37% orthophosphoric acid solution to increase the surface’s roughness and thus increase the shear bond strength of the material to the enamel. Currently, increasing studies are being performed to modify conventional techniques of enamel preparation to increase the adhesion efficiency [1,2]. The modification methods include enamel pre-etching with 32–37% orthophosphoric acid solution for 10–15 s, followed by using a self-etching bonding system. Another method leading to increased bonding properties is the prolonged (up to 120 s) application time of the etching agent [2].

Nowadays, various lasers are used in dental procedures as innovative therapeutic procedures that decrease bleeding, swelling, and pain [3,4,5,6,7]. The use of erbium family lasers has many advantages in conservative dentistry. These lasers allow cavity preparation (caries removal), caries prevention, a decrease of dentin hypersensitivity, decontaminate the tooth after caries removal, and are less painful during caries treatment [8]. In conservative dentistry, one of the alternative methods of modifying the enamel prismatic structure is using Er:YAG lasers [8]. By using erbium lasers, it is possible to prepare the tooth by creating the photothermal effect (vaporization) by heating up water contained in the tooth tissues. That leads to the tearing of the enamel and dentine structures, causing a secondary photomechanical effect. The photomechanical effect caused by lasers with wavelengths of 2940 nm (Er:YAG), 2780 nm (Er,Cr:YSGG) allows enamel preparation at different depths, depending on the values of such parameters like energy, frequency and exposure time. The preparation of tooth tissues with erbium lasers is very effective as it involves vaporization of water without excessive overheating of the irradiated tissues [9,10].

The key to obtaining a high shear bond strength of the composite material to a tooth is the proper preparation (conditioning) of the enamel surface. In 1975, Silvestrona et al. [11] distinguished three types of surfaces obtained after etching the enamel with the phosphoric acid solution of different concentrations (20–70%) placed on the enamel for 1 to 10 min. Subsequent studies on shear bond strength of restorative materials to various types of enamel have shown that the best adhesive properties are obtained for type I etching pattern, according to the classification of Silverstone, where the central regions of enamel prisms are dissolved without significant damage to the peripheral regions of prisms [12,13]. In type II of Silverstone classification, damage to enamel prisms is limited mainly to peripheral regions. As a result of the Er:YAG laser conditioning, the structure formed on the enamel surface is similar to Silverstone’s type III patterns (damage to peripheral and central regions of prisms) [14,15].

The study aimed to compare shear bond strength of orthodontic brackets to enamel and to provide a quantitative and qualitative assessment of the enamel surface, using X-ray microtomography of teeth conditioned with 37% orthophosphoric acid solution and Er:YAG laser, with or without additional etching of the enamel surface with 37% orthophosphoric acid solution. The null hypothesis in the study was that there are no differences in adhesion quality and roughness of the enamel after conditioning with Er:YAG laser or 37% orthophosphoric acid.

## 2. Materials and Methods

The research material consisted of healthy, noncarious, non-fractured, first premolars (n = 9) and third molars (n = 9) removed at the Dental Surgery Department of Wroclaw Medical University (Wrocław, Poland) for orthodontic indications. After the extraction, the teeth were stored in 1% of thymol solution in physiological saline to prevent developing bacterial flora. The teeth were stored for one month before the study was performed. The study was conducted in line with the approval no. KB 132/2019 issued by the Bioethics Committee appointed by the Rector of Wroclaw Medical University.

### 2.1. Enamel Surface Preparation

Group one (G1) were teeth with enamel conditioned using Er:YAG laser (LightTouch, LightInstruments, Yokneam, Israel) with the following operating parameters; energy: 100 mJ, frequency 10 Hz, exposure time: 10 s, applicator diameter 600 um, energy density: 35.37 J/cm^2^, distance: 1 mm away from a tooth, water spray cooling: 80%. Irradiation of the tooth surface, with an area similar to the size of the orthodontic bracket, was performed manually with S-shaped movements.

Group two (G2)—the samples were treated with 37% orthophosphoric acid solution (Arkona, Nasutów, Poland) for 30 s, and then the etchant was rinsed with distilled water for 10 s.

Group three (G3): the samples were conditioned using Er:YAG laser (LightTouch, LightInstruments, Yokneam, Israel) with the parameters identical to those for group one (G1) and, additionally, the samples were etched with 37% orthophosphoric acid solution for 15 s and then the etchant was rinsed with distilled water for 10 s.

### 2.2. Orthodontic Brackets Placement

Metal orthodontic brackets (GC Corp., Tokyo, Japan) were bonded to the samples by an orthodontic specialist, using Transbond XT LC material (3M Unitek, Seefeld, Germany) according to the following protocol: the surfaces of the enamel were polished using a brush Pro-Cup^®^ (Kerr, Brea, CA, USA) with non-fluoridated pumice paste Pressage (Shofu Inc., Kyoto, Japan) for 15 s, rinsed, and dried with air. Then, the teeth were etched with a 37% phosphoric acid—etching gel cobalt blue (Chemidental, Pabianice, Poland); for 30 s, rinsed with water and dried with compressed air. Adhesive primer Transbond™ XT LC (3M Unitek, Puchheim, Germany) was rubbed onto the enamel for 15 s. Adhesive material Transbond™ XT LC (3M Unitek, Neuss, Germany) was placed on the inner bracket’s surface, and then the bracket was placed in the middle of the mesial-distal axis of the tooth, moving its center 3.5 mm from the edge of the occlusal surface. The brackets were then exposed to 1200 W/cm^2^ polymerization lamp (Woodpecker, Nanning, China) and were irradiated from 4 sides (upper, lower, proximal and distal) for 20 s. The teeth were then stored in 1% thymol solution for one week before the measurement of the bracket debonding force was made, and the assessment of the enamel surface after the removal of the brackets was performed.

### 2.3. Shear Bond Strength Measurement

To determine the shear bond strength of orthodontic brackets to enamel, the shear bond strength test was performed. Each of the samples (a tooth with an orthodontic bracket) was placed in a polyethylene container using Duracryl^TM^ Plus self-polymerizing denture base resin (Spofa Dental, Jičín, Czech Republic) (Figure 1). Due to the shape and curvature of the tooth and the bracket, each sample was dipped in such a way that the punch could be placed at the front side, tangentially to the tooth surface.

To determine the adhesion value of an orthodontic bracket to the enamel, a shear bond strength test was performed using MTS 858 MiniBionix^®^ machine (MTS System, Eden Prairie, MN, USA). The measuring system is illustrated in Figure 2A,B. The load was applied to the sample using a punch, at a speed of 1 mm/min, tangentially to the side surface of the tooth and axially along the longer edge of the attached bracket [16]. (Figure 2C)

The shear bond strength value was determined by the formula [17]. The detailed results of the test are presented in Table 1 and Figure 3.
*τ* = *P_max_/b_z_z*
where *τ*—shear stress in the tooth-adhesive-bracket complex, *P_max_*—a value of maximum force obtained in the test, *b_z_*—width of the glue layer, *z*—height of the glue layer [17].

### 2.4. Assessment of the Enamel Surface after Conditioning

To assess the type of changes in the enamel after application of different conditioning techniques (Er:YAG laser, orthophosphoric acid, Er:YAG laser + orthophosphoric acid), the X-ray microtomography (SkyScan 1172, Bruker, Kontich, Belgium) was performed. Each sample was X-rayed, and the image was recorded using a resolution of 9 µm. The following parameters of the lamp were: 90 kV/112 µA and using Al and Cu filters. The exposure time was 1140 ms, the rotation angle of the support stand was 360͒, and the rotation step was 0.4͒͒. To be able to see the results of the conditioning, each sample was X-rayed twice—before and after the conditioning of the enamel. In this way, 2 images were obtained, which, when placed one on top of the other, allowed to see the elements of the tooth, which were modified (Figure 4).

The final stage of the research was to measure the values of parameters describing the roughness of the tooth’s enamel surface—before and after the conditioning. The measurement was performed using the DIAVITE DH-5 profilometer (Hahn and Kolb, Ludwigsburg, Germany). The length of the elementary segment was 0.5 mm, and the feed speed was v = 0.5 mm/s. Using the profilometer, the following surface roughness parameters were determined: Ra, Rz, Rmax, R3z, Rt, Rq. The measurement was performed for the frontal plane of the tooth.

### 2.5. Statistical Analysis

The mean values of the shear bond strength of the tooth-adhesive-bracket complex and the mean values of enamel roughness were compared with the variance analysis and post hoc tests (multiple comparisons using the Tukey’s test). The statistical analysis was conducted using Origin 5.0 software (OriginLab, Northampton, MA, USA). Values below *p* = 0.05 were considered to be statistically significant.

## 3. Results

### 3.1. The Results of the Shear Bond Strength Test

The samples in the G3 group were characterized by the highest shear stress values. The average value of shear stress is 9.28 MPa, while the lowest values were noticed for the G1 group, an average of 6.44 MPa (conditioned with Er:YAG laser).

### 3.2. Damage Mechanism of the Tooth-Adhesive-Bracket Complex

All tested samples were damaged due to the adhesive damage mechanism, i.e., loss of adhesion to bonded elements. This type of damage occurs when the external load exceeds the stress limit value of the adhesive bond to the materials that require bonding and have a much higher stiffness. Due to the bonding method and the materials that the tested sample is composed of the obtained shear strength values depend on the surface preparation (tooth enamel), which is related to its roughness, surface development and surface quality.

The analysis of the obtained results indicates that the lowest value of shear strength was typical for G1 group samples, for which the damage mechanism was related to the loosening of the adhesive and the orthodontic bracket from the enamel surface. Debonding of orthodontic brackets may occur at two boundaries: bracket-composite and enamel-composite interfaces. When the bond strength at the enamel-composite interface is greater than the bracket-composite interface, the bracket is removed, leaving adhesive material on the tooth’s surface. Whereas, during the opposite situation, the adhesive material is detached from the enamel. Furthermore, if the bond strength at both interfaces is over 10 MPa, debonding can damage the enamel by detaching the prisms during brackets removal [9]. The G3 group showed the highest stress values; some samples had stress values exceeding 10 MPa. Such a high-stress value, present at the tooth-adhesive-bracket complex, can cause enamel damage when the bracket is removed after treatment. However, the obtained values coincide with the results published in other studies [11,12]. In both the G2 group and G3 group, the damage mechanism of the tooth-adhesive-bracket complex was related to the loosening of the bracket from the adhesive surface, leaving the adhesive completely or partially on the enamel surface Figure 5.

### 3.3. The Analysis of the Enamel Surface, Prepared by Different Conditioning Techniques

The surface of tooth enamel treated with orthophosphoric acid solution and Er:YAG laser was analyzed using X-ray microtomography and roughness measurement. Figure 5 shows pictures for 1 of the example samples from each group. Each sample was analyzed twice: before and after surface preparation for the bracket. The presentation of surface differences was obtained by superimposing the obtained projections using the DataViewer software (SkyScan 1172, Bruker, Kontich, Belgium). The studies using X-ray microtomography clearly show damage to the enamel layer after laser application, regardless of G1 or G3 group. In each of these groups, the power value and laser time were the same for each sample. The analysis of the thickness of lesions in the enamel after laser treatment (using such set parameters) showed that the said lesions occurred at a depth of 0.01 to 0.09 mm. It means that in some regions, those lesions reached even 8–9% of the enamel thickness. Figure 6.

### 3.4. Roughness Parameters Were Measured on the Same Enamel Surfaces Prepared for Orthodontic Bracket Installation

The values of the roughness parameters were not only determined for individual groups (G1–G3) but also for the unconditioned enamel surfaces, which constituted the reference group. The detailed values of the parameters determining the enamel roughness are shown in Table 1. Based on the obtained results, it could be concluded that each of the conditioning techniques increased the roughness of the enamel surface prepared for the orthodontic bracket. The differences between the reference group and each study group were statistically significant (*p* < 0.05). The G2 group showed the smallest difference in roughness parameters (by 25% in the Ra parameter on average and 20% for Rz). The G1 group showed the highest difference (by more than 200% in both Ra and Rz parameter values). The G3 group, conditioned in two stages, also increased roughness values compared to the reference group. However, the said group also showed a decrease compared to the group conditioned with Er:YAG laser (by 16% for the Ra parameter and by 20% for the Rz parameter) (Table 2).

## 4. Discussion

One of the main problems that are still present in restorative dentistry is the achievement of an attachment between the composite materials and the enamel surface durable enough to maintain the long-term tightness of fillings and also reduces the risk of bacterial microleakage [8,10]. Composite bonds to the enamel surface as a result of shear bond strength that depends primarily on the type of the enamel surface, which can be modified in many ways. The results of the presented study proved that conditioning of enamel with an Er:YAG laser at 100 mJ/10 Hz led to greater changes of prismatic structures (increased roughness) compared to 37% orthophosphoric acid. Furthermore, enamel preparation using Er:YAG laser (100 mJ/10 Hz) combined with classic etching using 37% orthophosphoric acid helped to improve the adhesion of orthodontic brackets to the tooth enamel. The bond strength between the composite material and enamel depends on many factors, such as conditioning methods, etchant concentration, consistency, etching time, degree and method of polymerization of the bonding material [18,19].

Excellent adhesion is closely related to the enamel surface pattern obtained after enamel conditioning. The adhesive properties are related to the uniformly rough enamel surface in which the central regions of prisms are modified (damaged) to a greater extent than the peripheral regions of prisms. Such a surface enables a high absorption rate for materials that bond dental enamel to restorative materials [20,21,22]. The best adhesive properties are obtained for type I according to Silverstone’s classification, where the central regions of prisms of the enamel are dissolved without significant damage to the peripheral regions of prisms [12,13]. When the enamel is etched with 37% orthophosphoric acid, type I or II surfaces are most commonly obtained. Conditioning of the enamel with Er:YAG laser leads to the formation of a rough structure on the enamel surface. The said structure is similar to Silverstone’s type II and III patterns (damage only to peripheral regions of prisms or peripheral and central regions of prisms), which have much worse adhesive properties for composite materials [11,15,16]. The results obtained in this study also confirmed lower adhesion for samples conditioned with Er:YAG laser alone.

The use of erbium family lasers on hard dental tissues creates a highly irregular surface [8,15,16]. The ablative effect of erbium lasers is related to the high water absorption of the electromagnetic wave at approx. 3000 nm. In the enamel structure, the central regions of prisms contain more water. Hence initially, they are more efficiently vaporized compared to peripheral regions of prisms [23]. However, the micro-explosion phenomenon induced by the laser beam in the hard tissues of the tooth causes additional cracking and disintegration of the prismatic structure of the enamel, which increases its surface roughness and irregularity. Thus finally, the enamel is more similar to the Silvestrone type III pattern. Martinez-Insura et al.’s study showed that the adhesion of orthodontic brackets was better when the enamel was etched with 37% orthophosphoric acid solution than at laser preparation at 160–200 mJ/4 Hz [24]. In contrast, Lee et al. [25] found no statistically significant differences between the groups with enamel conditioned with 37% orthophosphoric acid solution and those conditioned with Er:YAG laser at 300 mJ/10 Hz. Interestingly, in contrast to the results of this study, the combined techniques showed a decrease in shear bond strength. Berk et al.’s [2] study showed that depending on used laser parameters (75–100 mJ/20 Hz), similar or different bond strength of the brackets to the enamel can be achieved compared to the etchant.

Laser conditioning can be used in many areas of dentistry. According to several studies, laser conditioning can improve the preservation of fissure sealants [26,27,28]. Various energy and frequency parameters were applied in the literature: 150 mJ/10 Hz [27], 60–100 mJ/2 Hz [28], 80 mJ/2 Hz [28]. The latter parameters resulted in good bonding properties of irradiated dental tissues [28]. Unlike the aforementioned studies, a combination of lower energy (100 mJ) and higher impulse frequency was used in the presented study, which, combined with orthophosphoric acid etching, ultimately affected the improvement of shear bond strength of orthodontic brackets to the dental enamel. Enamel conditioning was also performed when composite materials were used for the tooth restoration and the bonding of orthodontic brackets. Akhoundi [29], using 200 mJ/10 Hz parameters, proved the presence of irregular, amorphous enamel structures in the SEM analysis. The said structures, however, had gaps that could serve as retention for bonding materials. [29] The findings are consistent with Olivi’s [30] study. Olivi found that the best predictable enamel structures were obtained at lower laser parameters (80 mJ/10 Hz). Moritz [31], using parameters of 180 mJ/2 Hz, achieved a composite bond strength of 48 N, compared to 55 N when using 37% orthophosphoric acid and 60 N for kinetic preparation of corundum.

The results presented in this study show a significant increase in the values of enamel surface roughness after laser treatment (Table 1). Moreover, using X-ray microtomography allowed us to demonstrate the extent of enamel lesions as well as to estimate the depth of damage. The use of X-ray microtomography, compared to published studies using the SEM technique, allowed to show the entire region of the enamel where lesions took place [12,32]. Those lesions occurred both in samples functionalized with Er:YAG laser (group G1) and those additionally etched with a phosphoric acid solution (group G3). Studies concerning the mechanical properties of the enamel-adhesive-bond complex clearly showed that the samples conditioned with phosphoric acid solution and laser had the highest shear stress values. This value for a part of the samples within that group (G3) exceeded the limit of 10 MPa, which additionally strengthened the effect of enamel damage in the case of the removal of the orthodontic bracket. However, it is a desirable phenomenon when there is a need to permanently fix the material used for the restoration of a dental defect. The obtained results are consistent with those published by other authors [33,34]. It should be underlined that adhesion depends on many variables, including the topography of the tooth preparation and the level of bonding material viscosity. Therefore, it is suggested that the tooth surface’s roughness may change the wettability and the bonding quality of adhesive materials. Furthermore, higher roughness may lead to an adhesion decrease, especially when a high-level viscosity adhesive system is applied. Therefore, appropriate bonding systems should be used for the high roughness of tooth surfaces.

The studies using X-ray microtomography show damage to the enamel layer after using both Er:YAG laser itself and combined with an etcher. The analysis of the thickness of enamel lesions after laser treatment at 100 mJ/10 Hz showed that those lesions reach up to 8–9% of the enamel layer in some regions. The studies published in the literature do not conclusively answer whether the groups conditioned with the standard phosphoric acid technique are significantly different from those treated with Er:YAG laser. On one hand, the study by Gokcelik et al. [35] showed no statistically significant differences, while such a difference was observed in the article published by Hosseini et al. [33] In the presented publication, a comparison of shear stress values between groups showed statistically significant differences between the acid- and laser-conditioned group (G3) and the laser-conditioned group (G1). However, the results of shear stress values show no statistically significant changes between the laser-conditioned group (G1) and the phosphoric acid-functionalized group (G2), which is consistent with the results of other studies [32,35].

The authors of this article are aware of introduced limitations (one laser operating parameter, several samples, type of bonding material, a method of mounting the samples during the shear test), which were minimized at each testing stage. Therefore, further research should be conducted to, for example, determine the effect of different laser parameters on the extent and nature of enamel damage on a larger number of test samples.

## 5. Conclusions

The use of Er:YAG laser at 100 mJ/10 Hz (1Watt) combined with 37% orthophosphoric acid for enamel conditioning results in the increase in shear bond strength of orthodontic brackets to the tooth. The use of Er:YAG laser at operating settings applied in this study leads to a much higher degree of enamel damage compared to etching with 37% orthophosphoric acid. Furthermore, Er:YAG laser conditioning increases enamel roughness in contrast to 37% orthophosphoric acid etching.

## Figures and Tables

**Figure 1 materials-14-02093-f001:**
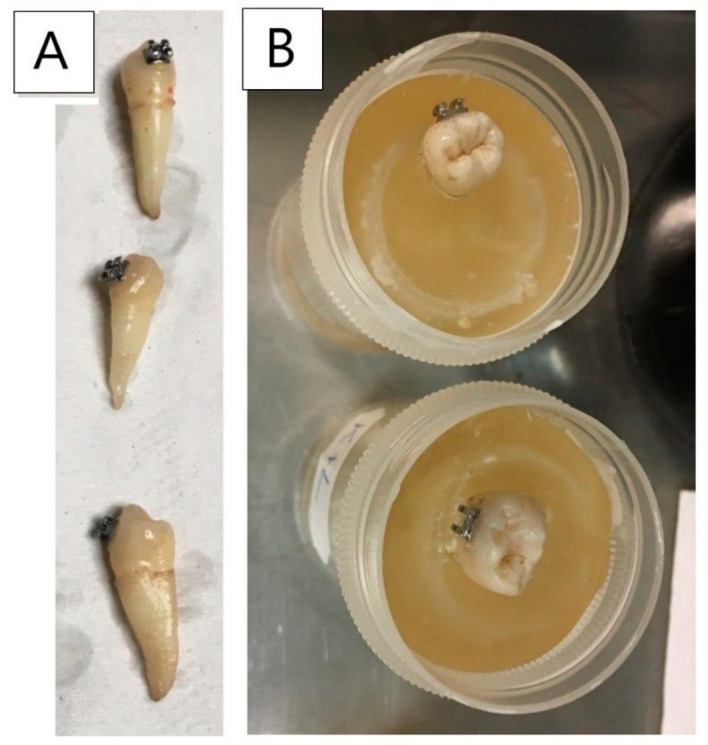
Samples: (**A**) teeth with an orthodontic bracket prepared for mechanical tests; (**B**) samples dipped in a polyethylene container using DuracrylTM Plus denture base resin (Spofa Dental, Jičín, Czech Republic).

**Figure 2 materials-14-02093-f002:**
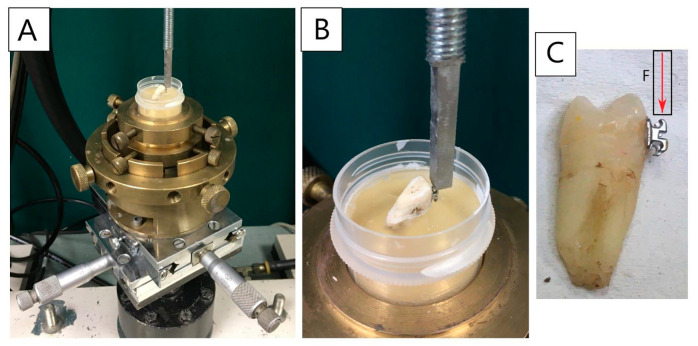
A working site prepared for the performance of shear bond strength test with a fixed sample and a picture of a tooth illustrating the direction of application of the load. (**A**,**B**) Method of placing the tooth in a measuring machine. (**C**) The direction of application of shear force.

**Figure 3 materials-14-02093-f003:**
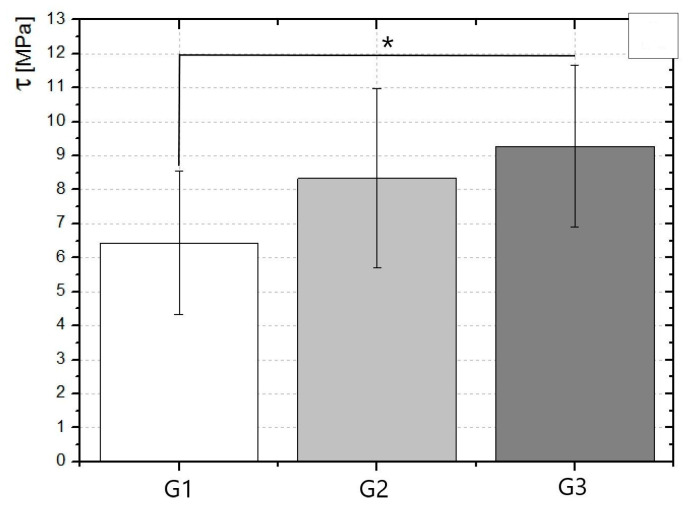
Comparison of average stress–strain values obtained in the shear bond strength test for 3 measurement groups: G1, G2 and G3, * *p* < 0.05.

**Figure 4 materials-14-02093-f004:**
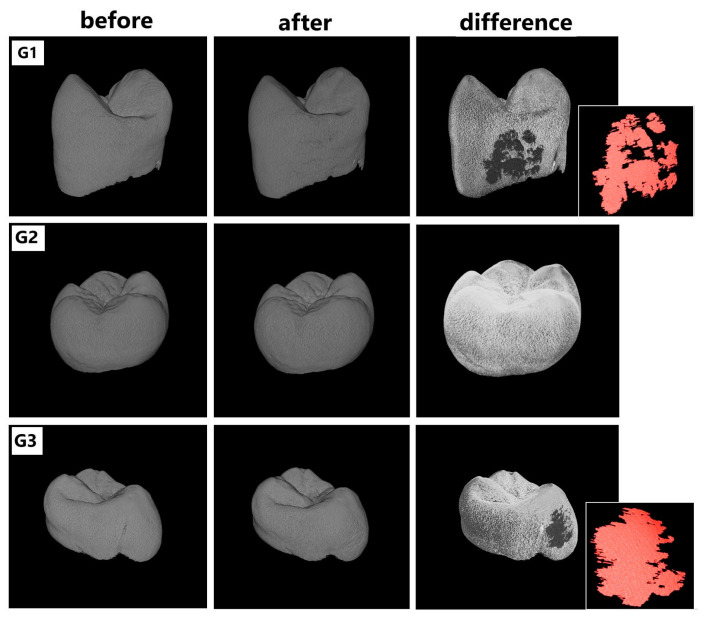
Comparison of pictures depicting the surface of example teeth of G1–G3 groups, conditioned with different techniques, taken using X-ray microtomography—1172 SkyScan, Bruker. The pictures depicting differences were obtained by subtractive operation using the DataViewer software (SkyScan 1172, Bruker, Kontich, Belgium).

**Figure 5 materials-14-02093-f005:**
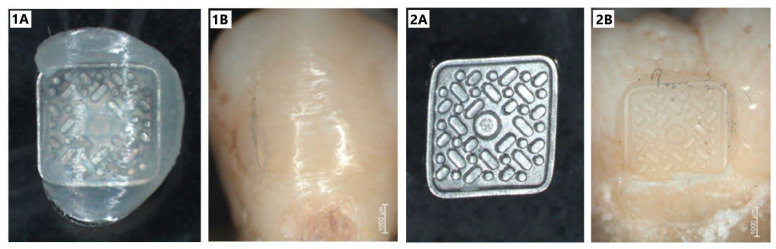
Comparison of examples of tooth surface and bracket surface with adhesive residue, adhesive damage of tooth-adhesive-bracket complex; (**1A**,**1B**) example surface of G1 group, conditioned with Er:YAG laser (τ = 6.38 MPa); (**2A**,**2B**) example surface of G3 group (τ = 10.13 MPa).

**Figure 6 materials-14-02093-f006:**
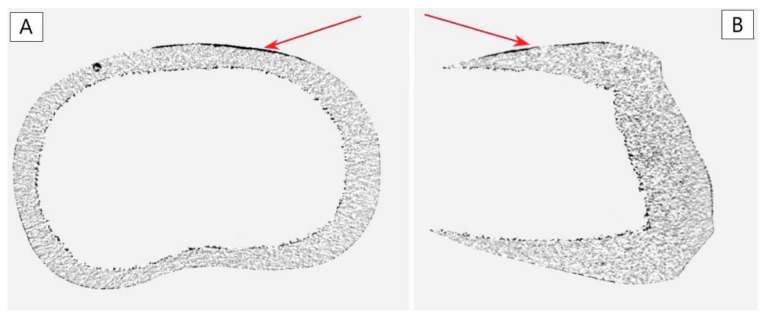
Example cross-sections (transverse and sagittal) of G1 group tooth, showing lesions in the enamel after laser conditioning (black). (**A**) Cross-section in the transverse plane; (**B**) cross-section in the sagittal plane. Pictures obtained using the DataViewer software (SkyScan 1172, Bruker, Kontich, Belgium).

**Table 1 materials-14-02093-t001:** Comparison of mean values of the shear bond strength, τ for the tooth-adhesive-bracket complex along with standard deviation values, minimum and maximum values, according to the study group; *p* < 0.05 using post hoc Tukey’s test.

	τ ([MP)	
Groups	Mean Value ± SD	Range (CI)
G1	6.44 ^a^ ± 2.11	3.84–9.75
G2	8.34 ^b^ ± 2.65	3.46–11.61
G3	9.28 ^a^ ± 2.38	4.97–12.09

^a^ Means with the same letter in the column show significant differences (*p* < 0.05), ^b^ means with the different letters in the column show no significant differences (*p* > 0.05).

**Table 2 materials-14-02093-t002:** Comparison of mean values of enamel roughness parameters, obtained for G1–G3 samples and enamel that were not treated with conditioning.

Groups	Ra (µm) Mean ± SD	Rz (µm) Mean ± SD	Rmax (µm) Mean ± SD	R3z (µm) Mean ± SD	Rt (µm) Mean ± SD	Rq (µm) Mean ± SD
**G1**	1.35 ^a,b,c^ ± 0.21	6.83 ^a,b,c^ ± 1.69	12.68 ^a,b^ ± 1.93	5.77 ^a,b,c^ ± 1.14	14.11 ^a,b^ ± 1.58	2.04 ^a,b,c^ ± 0.19
**G2**	0.56 ^a–d^ ± 0.07	2.23 ^a,d^ ± 0.53	4.2 ^a–d^ ± 1.49	1.76 ^a–d^ ± 0.48	5.42 ^a–d^ ± 0.88	0.88 ^a–d^ ± 0.18
**G3**	1.13 ^a,c,d^ ± 0.25	5.46 ^a,c,d^ ± 1.34	12.04 ^a,d^ ± 3.86	4.23 ^a,c,d^ ± 1.11	12.98 ^a,d^ ± 3.65	1.8 ^a,c,d^ ± 0.37
**Enamel**	0.44 ± 0.08	1.79 ± 0.5	3.56 ± 1.61	1.47 ± 0.36	4.1 ± 1.6	0.66 ± 0.15

^a^ comparison between individual groups and the reference group (enamel untreated with conditioning), *p* < 0.05; ^b^ comparison between G1 and G2 groups, *p* < 0.05, ^c^ comparison between G1 and G3 groups, *p* < 0.05; ^d^ comparison between G2 and G3 groups, *p* < 0.05.

## Data Availability

Not applicable.

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
