# Peer review of "The Effect of Er:YAG Laser on a Shear Bond Strength Value of Orthodontic Brackets to Enamel—A Preliminary Study"

_materials, 2021, doi:10.3390/ma14092093_

Round 1

Reviewer 1 Report

This paper presents finding of using Er:YAG laser pre-treatment combined with conventional etching can improve bonding strength. However, the conclusion of using only Er: YAG laser treatment should not be used for conditioning is lacking ground. The parameter space, including laser power, rep rate, exposure time, laser beam profile, etc., is actually quite big for optimizing surface morphology and increase bonding strength. 

Also, the question of why rougher surface lead to worse bonding strength is not very clear to the audience in the laser only case, while in the etching only case the surface is the smoothest but show a higher bonding strength than the laser only case. This will need more clarification.

Laser + etching show bonding strength of 9.28 +/- 2.38 MPa, and etching only show bonding strength of 8.34 +/- 2.65 MPa. The increase of the strength is marginal, and maybe within the error range. Understood the sample size is limited, so given such a small strength increase, whether it is true strength increase or it is within the experimental fluctuation error need more validation. 

The presentation need a little more polishing to bring it to a presentable format.   

Author Response

Dear reviewer, thank you for your great effort in review of  our manuscript.

Comments to the Author

This paper presents finding of using Er:YAG laser pre-treatment combined with conventional etching can improve bonding strength.

However, the conclusion of using only Er: YAG laser treatment should not be used for conditioning is lacking ground.

AD1) The conclusion was corrected, we removed the following sentence: “The study results indicate that Er:YAG laser should not be used for conditioning enamel without the additional use of any conventional etchant due to the profound damage to the prismatic structure of the enamel.”

The parameter space, including laser power, rep rate, exposure time, laser beam profile, etc., is actually quite big for optimizing surface morphology and increase bonding strength. 

Ad2) The study and the paper were prepared in the context of erbium laser use in conservative dentistry for tooth preparation. We aimed to check does is possible applying the laser alone or additional etching is demanded. Our literature review indicated studies suggest using the laser parameters of 80mJ/10Hz. Theoretically, to ablate the enamel, we need the fluence over 10-12J/cm2; in our study, theoretical fluence was 33J/cm2; however, due to the energy loss caused by the distance, the used fluence should be higher. We agree that the study using different energy densities should be done in future research. We added the parameters of the laser to the Abstract

Also, the question of why rougher surface lead to worse bonding strength is not very clear to the audience in the laser only case, while in the etching only case the surface is the smoothest but show a higher bonding strength than the laser only case. This will need more clarification.

Ad3) We added the following sentence to the Discussion: “It should be underlined that adhesion depends on many variables, including the topography of the tooth preparation and the level of bonding material viscosity. Therefore, it is suggested that the tooth surface's roughness may change the wettability and the bonding quality of adhesive materials. Furthermore, higher roughness may lead to an adhesion decrease, especially when a high-level viscosity adhesive system is applied. Therefore, appropriate bonding systems should be used for high rough tooth surfaces.”

Laser + etching shows a bonding strength of 9.28 +/- 2.38 MPa, and etching only shows a bonding strength of 8.34 +/- 2.65 MPa. The increase of the strength is marginal, and maybe within the error range. Understood the sample size is limited, so given such a small strength increase, whether it is true strength increase or it is within the experimental fluctuation error need more validation. 

Ad4) We understand that the differences between both groups are minimal, therefore we describe the limitation of the study (sample size) in the discussion section.

The presentation needs a little more polishing to bring it to a presentable format.   

Ad5)The language was polished by the English editor.

All the changes are highlighted in blue color.

Reviewer 2 Report

ABSTRACT

Line 18-19: please remove the manufacturer of the Er:YAG laser from the background section and move it to the Methods section (line 21). Or, if you prefer, add in the background section a sentence like the following: “the aim of the present study is to evaluate the effect of an Er:YAG laser (LightTouch, LightInstruments, Israel) on…..”

Line 21: please specify in brackets the parameters of the laser (100mJ/10Hz).

Line 27: please replace “. p<0.05” with “(p<0.05).”

Throughout all the manuscript please add references before the point. E.g. “[1].” Instead of “.[1]”

INTRODUCTION

Line 75-83: You state that enamel conditioning with Er:YAG laser causes much worse adhesive properties for composite materials. If so, what is the rationale of your study? Please specify.

At the end of the introduction please clearly state the statistical null hypotheses.

MATERIALS AND METHODS

Why different types of teeth have been tested (premolars and molars)? Might the specific tooth influence the parameters tested?

How was the number of samples established? Did you perform a sample size calculation?

Were teeth excluded in some cases? Did you consider exclusion criteria? Teeth should have been observed with a stereomicroscope and excluded in case of caries, fractures, etc.

Might the one month wait in solution before proceeding with the experimentation have altered the enamel characteristics of the teeth?

Line 129: please add a reference for the punch speed of 1 mm/min. Refer for example to the following: Scribante A, Gallo S, Turcato B, Trovati F, Gandini P, Sfondrini MF. Fear of the Relapse: Effect of Composite Type on Adhesion Efficacy of Upper and Lower Orthodontic Fixed Retainers: In Vitro Investigation and Randomized Clinical Trial. Polymers (Basel). 2020 Apr 21;12(4):963. doi: 10.3390/polym12040963. PMID: 32326201; PMCID: PMC7240513.

Please add a reference for the formula used for the shear bond strength.

Besides the parameters tested, you should evaluate even the adhesive remnant index after shear bond strength. In order to do so, you could assess the ARI index or better by means of MicroCT scans. As well, enamel roughness should be evaluated even after brackets detachment.

At the end of the materials and methods, please specify the statistical analyses.

RESULTS

Results should be better presented.

Table 1: please add “G2” instead of “2” in the first column. Moreover, in the second column where you present the means, you should add the same superscript letter if there are no significant differences between the groups, whereas a different letter in case of significant difference. For instance, add “a, b, b” respectively for G1, G2, and G3. Moreover, at the end of the table add the following footnote: “means with the same letter show no significant differences (p>0.05)”

I actually don’t completely understand figure 3B. please specify.

Line 161-163: you state that G1 was conditioned with Er:YAG laser. But in the abstract you state that it was G2. Please clearly define the groups in the entire manuscript.

Line 233: this must be entitled “Table 2”. The results are presented with confusion. Please improve the quality presentation.

DISCUSSION

Line 269-271: this sentence seems to contrast with the previous where you stated that Er:YAG laser generally cause Silverstone’s type II and III patterns.

Author Response

Dear reviewer, thank you for your great effort in the review of our manuscript.

Comments to the Author

ABSTRACT

Line 18-19: please remove the manufacturer of the Er:YAG laser from the background section and move it to the Methods section (line 21). Or, if you prefer, add in the background section a sentence like the following: “the aim of the present study is to evaluate the effect of an Er:YAG laser (LightTouch, LightInstruments, Israel) on…..”

Ad) We changed the abstract's background in accordance with the reviewer's remark.

Line 21: please specify in brackets the parameters of the laser (100mJ/10Hz).

Line 27: please replace “. p<0.05” with “(p<0.05).”

Ad) The parameters of the laser and “line 27” were changed.

Throughout all the manuscript please add references before the point. E.g. “[1].” Instead of “.[1]”

Ad) We changed the paper in accordance with the reviewer's remark.

INTRODUCTION

Line 75-83: You state that enamel conditioning with Er:YAG laser causes much worse adhesive properties for composite materials. If so, what is the rationale of your study? Please specify.

Ad) The sentence “with much worse adhesive properties for composite materials” was removed from the Introduction.

At the end of the introduction please clearly state the statistical null hypotheses.

Ad) We add the sentence to the introduction: “The null hypothesis in the study was there are no differences in adhesion quality and roughness of the enamel after conditioning with Er:YAG laser or 37% ortho-phosphoric acid.”

MATERIALS AND METHODS

Why different types of teeth have been tested (premolars and molars)? Might the specific tooth influence the parameters tested?

AD) Prior to the experiment, we did preliminary studies where we found no differences in enamel pattern and shear bond strength of orthodontics brackets between different teeth types (canines, premolars, molars) for the same type of brackets. Thank you for this very important remark.

How was the number of samples established? Did you perform a sample size calculation?

Ad) The sample size calculation was not done due to the limited sample size. We changed the title of the paper to “Effect of Er:YAG laser on a shear bond strength value of orthodontic brackets to enamel – preliminary study” Limitation of the study was also described in the discussion.

Were teeth excluded in some cases? Did you consider exclusion criteria? Teeth should have been observed with a stereomicroscope and excluded in case of caries, fractures, etc.

Ad) All the teeth were extracted due to orthodontics reasons. Teeth were without fracture or caries lesions. We added the data to M&M section.

Might the one month wait in solution before proceeding with the experimentation have altered the enamel characteristics of the teeth?

Ad) Our experience and previous studies showed that this type of teeth storage does not alter the enamel when compared with fresh teeth.  Also, a similar storage procedure is recommended by some other researchers.

Line 129: please add a reference for the punch speed of 1 mm/min. Refer for example to the following: Scribante A, Gallo S, Turcato B, Trovati F, Gandini P, Sfondrini MF. Fear of the Relapse: Effect of Composite Type on Adhesion Efficacy of Upper and Lower Orthodontic Fixed Retainers: In Vitro Investigation and Randomized Clinical Trial. Polymers (Basel). 2020 Apr 21;12(4):963. doi: 10.3390/polym12040963. PMID: 32326201; PMCID: PMC7240513.

Ad) The reference was added.

Please add a reference for the formula used for the shear bond strength.

Ad) The reference was added.

Besides the parameters tested, you should evaluate even the adhesive remnant index after shear bond strength. In order to do so, you could assess the ARI index or better by means of MicroCT scans. As well, enamel roughness should be evaluated even after brackets detachment.

AD) Thank you for your important suggestion. We will add the ARI score in our future research we will do.

At the end of the materials and methods, please specify the statistical analyses.

AD) Statistical Analysis was added to the M&M section

RESULTS

Results should be better presented.

Table 1: please add “G2” instead of “2” in the first column. Moreover, in the second column where you present the means, you should add the same superscript letter if there are no significant differences between the groups, whereas a different letter in case of significant difference. For instance, add “a, b, b” respectively for G1, G2, and G3. Moreover, at the end of the table add the following footnote: “means with the same letter show no significant differences (p>0.05)”

AD) Table 1 was modified.

I actually don’t completely understand figure 3B. please specify.

AD) We decided to remove the Figure 3B

Line 161-163: you state that G1 was conditioned with Er:YAG laser. But in the abstract, you state that it was G2. Please clearly define the groups in the entire manuscript.

Ad) The mistake on the abstract was corrected. (The groups in the study were: G1-laser, G2-acid etching, G3-Laser+acid etching)

Line 233: this must be entitled “Table 2”. The results are presented with confusion. Please improve the quality presentation.

AD) Table 2 was modified as recommended.

DISCUSSION

Line 269-271: this sentence seems to contrast with the previous where you stated that Er:YAG laser generally causes Silverstone’s type II and III patterns.

AD) The incorrect sentence was corrected in the introduction to „In type II of Silverstone classification damage of enamel prisms is limited mainly to peripheral regions. As a result of enamel conditioning with Er:YAG laser, the structure is formed on the enamel surface which is similar to Silverstone's type III patterns (damage to peripheral and central regions of prisms)“

and in the discussion to

„The ablative effect of erbium lasers is related to the high absorption of the electromagnetic wave at approx. 3,000 nm by water. In the enamel structure, the central regions of prisms contain more water, hence initially they are more efficiently vaporized compared to peripheral regions of prisms[23]. However, the micro-explosion phenomenon induced by the laser beam in the hard tissues of the tooth causes additional cracking and disintegration of the prismatic structure of the enamel, which increases its surface roughness and irregularity, thus finally the enamel is more similar to the Silvestrone type III pattern.“

The changes are highlighted in blue color.

Reviewer 3 Report

Please draft abstract as per materials journal requirements !

The entire work should be checked for type and some English poor formulation

This is not clear “leads to a much higher degree of enamel damage compared to etching” in the actual formulation I understand that “The use of Er:YAG laser” is detrimental

“Furthermore, the use of erbium lasers reduces the risk of bacterial leakage which may lead to secondary caries” this is wrong statement

Overall the introduction is very poor structured and also difficult to understand if the use of “Er:YAG laser” is beneficial or not !!

You said “The research material consisted of human molars and premolars (n=18),” however in introduction you refer to “The research material consisted of healthy first premolars (n=9)”

Please provide details “to the protocol recommended by the manufacturer” how to replicate it without details

Why was sued this “at a speed of 1mm/min”  it correspond to any standard or physical value ?

The font size and legend in Figure 3 is very small please increase to easy interpret the values. For example in Figure 3 b is almost impossible to distinguish what are the values in X and Y axes

The damage mechanism is very poorly presented , it looks rather  report nota  research discussion, for example line starting with 188 is just a stamen not any details of interfacial fracture or so on ..

The best part seems the discussion, other should eb checked in details cause are many parts that contradicts ..also the conclusions do not agree with discussions section

Author Response

Dear reviewer, thank you for your great effort in the review of our manuscript.

Comments to the Author
Please draft abstract as per materials journal requirements !

Ad) The abstract was modified according to journal requirements.

The entire work should be checked for type and some English poor formulation

Ad) The paper was checked by an English editor.

This is not clear “leads to a much higher degree of enamel damage compared to etching” in the actual formulation I understand that “The use of Er:YAG laser” is detrimental

Ad) The use of Er:YAG laser has many advantages in conservative dentistry, e.g., lack of vibration, is less painful and allows to decontaminate the tooth after caries removal. However, we wanted to check whether the enamel's prismatic structure's etching is mandatory and does conventional etching can be replaced by the laser itself. The study results showed that laser conditioning alone is not recommended and the enamel after laser preparation (clinically when caries removal) should be etching by conventional acids. Additional etching of the lased enamel is beneficial and increases the adhesion. We modified the conclusion to underline the benefit of Er:YAG laser enamel conditioning combined with conventional etching in contrast to etching alone.

“Furthermore, the use of erbium lasers reduces the risk of bacterial leakage which may lead to secondary caries” this is wrong statement

Ad) The sentence was removed.

Overall the introduction is very poor structured and also difficult to understand if the use of “Er:YAG laser” is beneficial or not !!

Ad) The introduction was rewritten

You said “The research material consisted of human molars and premolars (n=18),” however in introduction you refer to “The research material consisted of healthy first premolars (n=9)”

Ad) The introduction was rewritten

Please provide details “to the protocol recommended by the manufacturer” how to replicate it without details

AD) We described the bonding protocol in the M&M section (2. Orthodontic brackets placement)

Why was sued this “at a speed of 1mm/min”  it correspond to any standard or physical value ?

AD) We added a reference for the punch speed of 1 mm/min: Scribante A, Gallo S, Turcato B, Trovati F, Gandini P, Sfondrini MF. Fear of the Relapse: Effect of Composite Type on Adhesion Efficacy of Upper and Lower Orthodontic Fixed Retainers: In Vitro Investigation and Randomized Clinical Trial. Polymers (Basel). 2020 Apr 21;12(4):963. doi: 10.3390/polym12040963. PMID: 32326201; PMCID: PMC7240513.

The font size and legend in Figure 3 is very small please increase to easy interpret the values. For example in Figure 3 b is almost impossible to distinguish what are the values in X and Y axes

AD) We decided to remove the Figure 3B

The damage mechanism is very poorly presented , it looks rather  report nota  research discussion, for example line starting with 188 is just a stamen not any details of interfacial fracture or so on ..

AD) We described the damage mechanism between bracket-adhesive and enamel-adhesive interfaces more deeply.

The best part seems the discussion, other should eb checked in details cause are many parts that contradicts ..also the conclusions do not agree with discussions section

Ad) The discussion was rewritten

The changes are highlighted in blue color.

Round 2

Reviewer 1 Report

"The ablative effect is related to high water absorption..." This is very confusing, why higher water leads to more ablation? any reference?

Author Response

Reviewer: 1

At the end of the materials and methods, please specify the statistical analyses. Comments and Suggestions for Authors

"The ablative effect is related to high water absorption..." This is very confusing, why higher water leads to more ablation? any reference?

Dear reviewer, thank you for your great effort in the review of our manuscript.

Ad) The Er:YAG laser has a wavelength of 2940 nm with coincides with the peak of absorption in the water.  The more water is in the tissue, the higher effectiveness of tissue ablation is. For example, enamel needs more energy to be vaporized (absorption threshold of 10-12 J/cm2, percentage of water 2-4%), dentin needs less energy to be evaporated (absorption threshold of 4 J/cm2, rate of water 20-30%)

References: https://www.laserandhealthacademy.com/media/objave/academy/priponke/17_22___lukac___water_absorption_shift___jlaha_2013_1.pdf

Reviewer 2 Report

Thanks for improving your manuscript according to my comments.

I just ask you to change "background" with "aim" in the abstract. Thank you.

Author Response

Comments and Suggestions for Authors

Thanks for improving your manuscript according to my comments.

I just ask you to change "background" with "aim" in the abstract. Thank you.

Dear reviewer, thank you for your great effort in the review of our manuscript.

AD) The abstract was prepared according to the Instruction to authors, that's why we used Background instead of the Aim.

„ 1) Background: Place the question addressed in a broad context and highlight the purpose of the study; 2) Methods: Describe briefly the main methods or treatments applied. Include any relevant preregistration numbers, and species and strains of any animals used. 3) Results: Summarize the article's main findings; and 4) Conclusion: Indicate the main conclusions or interpretations.“

Reviewer 3 Report

The authors have well responded to my concerns therefore I suggest acceptance.

Author Response

Reviewer: 3

Comments and Suggestions for Authors

The authors have well responded to my concerns therefore I suggest acceptance.

Ad) Dear reviewer, thank you for your great effort in the review of our manuscript.